# Indian Ocean variability changes in the Paleoclimate Modelling Intercomparison Project

**Chris Brierley**[1]**, Kaustubh Thirumalai**[2]**, Edward Grindrod**[1]**, and Jonathan Barnsley**[1]

[1]Dept. Geography, University College London, London, United Kingdom
[2]Dept. Geosciences, University of Arizona, Tucson, Arizona, USA

**Correspondence:** Chris Brierley (c.brierley@ucl.ac.uk)

**Abstract.** The Indian Ocean exhibits multiple modes of interannual climate variability, whose future behaviour is uncertain. Recent analysis of glacial climates has uncovered an additional El Niño-like equatorial mode in the Indian Ocean, which could also emerge in future warm states. Here we explore changes in the tropical Indian Ocean simulated by the Paleoclimate Model Intercomparison Project (PMIP4). These simulations are performed by an ensemble of models contributing to the Coupled Model Intercomparison Project 6 and over four coordinated experiments: three past periods – the mid-Holocene (6000 years ago), the Last Glacial Maximum (21 000 years ago), the last interglacial (127 000 years ago) – and an idealized forcing scenario to examine the impact of greenhouse forcing. The two interglacial experiments are used to characterize the role of orbital variations in the seasonal cycle, whilst the other pair focus on responses to large changes in global temperature.

The Indian Ocean Basin Mode (IOBM) is damped in both the mid-Holocene and last interglacial, with the amount related to the damping of the El Niño–Southern Oscillation in the Pacific. No coherent changes in the strength of the IOBM are seen with global temperature changes; neither are changes in the Indian Ocean Dipole (IOD) nor the Niño-like mode. Under orbital forcing, the IOD robustly weakens during the mid-Holocene experiment, with only minor reductions in amplitude during the last interglacial. Orbital changes do impact the SST pattern of the Indian Ocean Dipole, with the cold pole reaching up to the Equator and extending along it. Induced changes in the regional seasonality are hypothesized to be an important control on changes in the Indian Ocean variability.

## 1 Introduction

The Indian Ocean (IO) is Earth's third largest ocean and unique amongst the tropical oceans as the Indo-Pacific Warm Pool (Wyrtki, 1989) and Maritime Continent prevent equatorial easterly winds and eastern upwelling that characterize both the tropical Pacific and Atlantic (Schott et al., 2009; Huang et al., 2016). Observations show the thermocline is almost flat between African and Indonesia (Cowan et al., 2015) due to the weak mean equatorial winds regulated by the monsoon climate (Xu et al., 2021). The Asian landmass inhibits meridional heat transport to the midlatitudes, prevents ventilation of the thermocline, and helps drive the Indian monsoon.

The Indian Ocean harbours many modes of interannual variability (e.g. Behera and Yamagata, 2001; Zhang et al., 2020), two of which were assessed by the IPCC (Cassou et al., 2021): the Indian Ocean Dipole (IOD; Saji et al., 1999; Webster et al., 1999) and the Indian Ocean Basin Mode (IOBM; Huang et al., 2016). These modes have global teleconnections but particularly pronounced effects around the densely populated IO rim, as demonstrated by the 2019 extreme IOD (Wainwright et al., 2021; Wang and Cai, 2020). Since 1950, the Indian Ocean basin has experienced rapid sea surface temperature (SST) warming, attributed to anthropogenic emissions (Gulev et al., 2021). The changing mean state will substantially modify the processes controlling the modes of variability.

## 1.1 Indian Ocean Dipole

The primary Indian Ocean mode is the IOD, discovered following a strong positive event in 1997 (Saji et al., 1999; Webster et al., 1999). The magnitude of IOD events is commonly measured using the Dipole Mode Index (DMI; see Sect. 2.3 for definition). The majority of IOD events are triggered by the El Niño–Southern Oscillation (ENSO; Stuecker et al., 2017). During an El Niño, the atmospheric convection in the Walker circulation shifts eastward, forcing anomalous subsidence over the Maritime Continent (Schott et al., 2009) and easterly wind anomalies along the equatorial IO (Liu et al., 2014), triggering a positive IOD event. Alternatively, some IOD events are initiated without any Pacific influence (Behera et al., 2006; Wang et al., 2016). Once SST anomalies form, regardless of their origin, several positive feedbacks sustain and enhance the IOD (Liu et al., 2014). As the zonal SST gradient initially decreases, the seasonal easterly winds strengthen, shoaling the thermocline and resulting in cold upwelling, which reinforces the initial cold anomaly (McKenna et al., 2020). Anomalous Ekman pumping warms the western Indian Ocean (Cai et al., 2021b). IOD events are characteristically phased-locked to begin developing around June, peak in boreal fall, and rapidly decay following the reversal of the trade winds in November (Schott et al., 2009; Abram et al., 2020a). The IOD has a periodicity of 3–5 years (Ashok et al., 2003; McKenna et al., 2020) with positive events being more intense than negative events (Cai et al., 2013). This skewness is mainly attributed to asymmetry in the thermocline–SST feedback, which overpowers increased damping from the SST–cloud–radiation feedback (Ng et al., 2014; Ogata et al., 2013).

Projected future changes in the Indian Ocean's mean state will have significant impacts on the IOD by modulating the efficiency of the feedbacks that control its evolution (Cai et al., 2013; Wang et al., 2021). Shoaling of the equatorial thermocline near Indonesia will strengthen the thermocline–SST feedback, encouraging stronger IOD events (Zheng et al., 2013; Abram et al., 2020a). On the other hand, increased atmospheric static stability will weaken the atmospheric components of the Bjerknes feedback (Cai et al., 2013). The changing feedbacks will modify the characteristics of the IOD, reducing skewness (Cowan et al., 2015) and shifting events towards boreal summer (Zheng et al., 2021). Although the frequency of positive IOD events has been rising, the consensus on the future frequency and strength of IOD is weak (Eyring et al., 2021). The overall frequency of IOD events is not projected to increase (Cai et al., 2013), yet proxies suggest past positive IOD-like mean states have experienced more frequent IODs (Abram et al., 2020a). Some studies find no increase in IOD magnitude (Zheng et al., 2013; Hui and Zheng, 2018) under future warming scenarios, whilst others have (Marathe et al., 2021). The projected frequency of extreme positive IOD events increases substantially as the zonal SST gradient and winds weaken (Cai et al.,

2014). However, an opposite response may occur for moderate positive events as tropospheric warming limits Ekman pumping (Cai et al., 2021b).

Investigations of past interannual variations in the Indian Ocean have facilitated insights into the nature of coupling between altered mean states and climate variability in this basin. Proxy reconstructions reveal elevated IOD variability during periods when the zonal SST gradient was reduced relative to present-day conditions, accompanied by enhanced mean cooling in the eastern Indian Ocean (see Abram et al., 2020a for a review). Such periods of intensified IOD activity alongside eastern cooling have been uncovered during the last millennium (Abram et al., 2020b), the mid-Holocene period (Abram et al., 2007), and the Last Glacial Maximum (LGM) (Thirumalai et al., 2019). Various lines of proxy evidence reinforce the finding that the Indian Ocean is capable of harbouring larger interannual climate variations than observed today (Abram et al., 2020a). Coeval records from the Pacific point to mean-state-dependent feedbacks maintaining pan-tropical IOD and ENSO interactions over different palaeoclimate periods. For example, strong IOD activity recorded over the last millennium appears to be tightly linked to strengthened ENSO variability as well (Abram et al., 2020b). During the mid-Holocene and the LGM however, when global climate and forcing were drastically altered relative to the last millennium, reconstructions point to stronger interannual variability (Thirumalai et al., 2019; Abram et al., 2007, 2020a) and weakened ENSO-related variability in the Pacific (Leduc et al., 2009; Ford et al., 2015; Abram et al., 2007). Though the ENSO–IOD relationship may depend on the palaeoclimate mean state and boundary conditions therein, proxy records support a tight link between the zonal SST gradient and IOD intensity (Abram et al., 2020a), with robust evidence for variations larger than those observed in the instrumental period.

There have been several previous studies looking at the Indian Ocean Dipole using palaeoclimate simulations by a single model. Brown et al. (2009) and Liu et al. (2007) use the FOAM model and find a slight weakening in IOD amplitude at the mid-Holocene and little change at the LGM. Meanwhile Iwakiri and Watanabe (2019) find a stronger IOD in the mid-Holocene in MIROC5. Rehfeld et al. (2020) showed that changes in IOD amplitude do not scale global mean temperature changes across a multi-model ensemble including palaeoclimate simulations. Combined, though, the literature does suggest IOD changes with changes in the seasonal cycle; something supported by the emerging, independent research of Liu et al. (2023), who look at the mechanisms behind this response.

## 1.2 Indian Ocean Basin Mode

The Indian Ocean Basin Mode (IOBM) is a basin-wide anomaly that is the leading mode of interannual Indian Ocean SST variability. IOBM events often follow IOD events, de-

veloping in boreal winter and peaking in the following spring (Wang, 2019). This seasonality implies a connection with the ENSO mature phase in boreal winter (Xu et al., 2021), although modelling suggests IOBM can occur without ENSO (Kajtar et al., 2017). Nonetheless, ENSO is the dominant forcing mechanism for the IOBM (Zhang et al., 2021) as an El Niño induces anomalous Walker circulation over the equatorial Indo-Pacific (Guo et al., 2018). This modulates surface winds, reducing evaporation and increasing downward short-wave radiation to warm the Indian Ocean (Wang, 2019). Diffusion of tropospheric temperature anomalies from the eastern Pacific contributes significantly (Tao et al., 2016). The IOBM–ENSO relationship displays interdecadal variation (Tao et al., 2015) with strong ENSO elevating the relationship as the thermocline shoals in the south-west Indian Ocean (Kajtar et al., 2017). Most Coupled Model Intercomparison Project (CMIP) 5 models capture the key features of the IOBM whilst overestimating its amplitude (Marathe et al., 2021), due to winter rainfall and thermocline biases (Tao et al., 2016). Initial investigations of the CMIP6 ensemble indicate underestimated SST variability results from inaccurate latent heat fluxes and wind-driven ocean processes (Halder et al., 2021).

IOBM future projections must consider the response and uncertainty of ENSO simulation (Collins et al., 2010). Under warming, the tropical Pacific moves towards an El Niño-like mean state (Vecchi and Soden, 2007) and a consensus is developing that ENSO-induced precipitation variability will intensify, canonical El Niño will strengthen, and both El Niño Modoki and extreme ENSO events will increase in frequency (Cai et al., 2021a; Stevenson et al., 2021). These changes will exacerbate the future global impacts of ENSO (Power and Delage, 2018) and should drive a stronger IOBM (Zheng, 2019). Indeed, many modelling studies suggest an enhanced IOBM and capacitor effect under global warming (Tao et al., 2016). This has been found alongside ENSO activity that is reduced (Zheng et al., 2011) or unchanged (Hu et al., 2014; Tao et al., 2015). The increased IOBM response results from strengthened air–sea interactions (Zheng et al., 2011) and a greater tropospheric temperature response (Hu et al., 2014; Tao et al., 2015). A recent study projects a weaker IOBM response in early summer suggesting a decreased capacitor effect, yet the IOBM feedback on ENSO transition is strengthened (Marathe et al., 2021). The IOBM is also projected to have an increasing influence over the East Asian summer climate (Huang et al., 2016; Qu and Huang, 2012).

Few studies have focused on reconstructing past IOBM variations. Complexities arise in assessing past IOBM behaviour due to the potentially confounding influence of the IOD, where warming in the east of the basin associated with negative IOD events could be conflated with anomalous IOBM warming (Abram et al., 2020b). Moreover, the lack of available interannually resolved records from the western Indian Ocean presents an additional challenge in isolating past IOBM intensity.

## 2 Methods

### 2.1 Models

Climate models, with their basis in geophysical fluid dynamics, are our best source of information about the future changes in climate. The vehicle that coordinates and collates simulations of future climate is the Coupled Model Intercomparison Project (CMIP) (Eyring et al., 2016b). The aspect of CMIP that looks at simulations of past is the Paleoclimate Model Intercomparison Project (PMIP). Here, we combine simulations from both PMIP phase 4 (PMIP4; Kageyama et al., 2018) and phase 3 (PMIP3) (Braconnot et al., 2012). This acts to increase the ensemble size (Brown et al., 2020) and the two phases have shown to be statistically indistinguishable (Brierley et al., 2020).

Models must have completed one or more palaeoclimate simulations to be included here and to have provided monthly surface temperature and precipitation for at least 30 years for both this simulation and the preindustrial control (see Sect. 2.2). The resulting 34 models are listed in Table 1, with a total of 34 426 simulated years analysed. Further information about the CMIP6 models is available in on the PMIP4 website (https://pmip4.lsce.ipsl.fr/doku.php/database:participants, last access: 7 July 2022), whilst details of the CMIP5 models can be found in Table 9.A.1 of Flato et al. (2013).

### 2.2 Experiments

This research uses simulations run under five different experiments defined under either the CMIP or PMIP protocols. All models have performed a preindustrial control (henceforth piControl) which approximates to constant 1850 CE boundary conditions (Eyring et al., 2016a). This simulation acts as a baseline from which changes are computed under all the other experiments. The piControl simulations vary in length but are a minimum of 100 years long (Table 1). The "abrupt4xCO2" experiment is another of the required deck of CMIP simulations and is an idealized forcing experiment where the carbon dioxide concentrations are instantaneously quadrupled and then the model is left to respond to them. These simulations are useful in estimating a model's climate sensitivity (Gregory et al., 2004; Zelinka et al., 2020), and the resulting values are shown in Table 1. Here we use the average of years 101–150 after the forcing change to provide an idea of climate changes in response to increasing greenhouse gases. This experiment and a segment of it are selected over other options as they are (a) unchanged between CMIP5 and CMIP6, (b) available for a large number of models, and (c) closer to equilibrium than the alternative "1pctCO2" experiment. Efforts are made to account for any transience in this segment through linear detrending (see Sect. 2.3).

The "midHolocene" experiment is the most simulated of all PMIP experiments, having both been part of PMIP since

**Table 1.** The ensembles analysed in this work. For each of the models, we provide the amount of simulation output analysed for each experiment (in years). Not all models performed all experiments. The "genre" shows whether a model produced future simulations as part of either CMIP5 or CMIP6 or only performed the palaeoclimate experiments (PMIP3 or PMIP4). Also shown is the equilibrium climate sensitivity (ECS, in $°C\,Wm^2$) from either Collins et al. (2013); Zelinka et al. (2020) or the article introducing the model.

| Model | Genre | ECS | piControl | midHolocene | lig127k | lgm | abrupt4xCO2 |
|---|---|---|---|---|---|---|---|
| ACCESS-ESM1-5 | CMIP6 | 3.9 | 900 | – | 200 | – | 150 |
| AWI-ESM-1-1-LR | CMIP6 | 3.6 | 100 | 100 | 100 | 100 | – |
| BCC-CSM1-1 | CMIP5 | 3.1 | 500 | 100 | – | – | 150 |
| CCSM4 | CMIP5 | 2.9 | 1051 | 301 | – | 101 | 150 |
| CESM2 | CMIP6 | 5.3 | 500 | 700 | 700 | – | 149 |
| CNRM-CM5 | CMIP5 | 3.3 | 850 | 200 | – | 200 | 150 |
| CNRM-CM6-1 | CMIP6 | 5.1 | 500 | – | 301 | – | 150 |
| COSMOS-ASO | PMIP3 | 4.7 | 400 | – | – | 600 | – |
| CSIRO-Mk3-6-0 | CMIP5 | 4.1 | 500 | 100 | – | – | 149 |
| CSIRO-Mk3L-1-2 | PMIP3 | 3.1 | 1000 | 500 | – | – | – |
| EC-Earth3-LR | CMIP6 | 4.3 | 201 | 201 | – | – | – |
| FGOALS-f3-L | CMIP6 | 3 | 561 | 500 | 500 | – | 150 |
| FGOALS-g2 | CMIP5 | 3.7 | 700 | 680 | – | 100 | 150 |
| FGOALS-g3 | CMIP6 | 2.9 | 500 | 500 | 500 | – | – |
| FGOALS-s2 | CMIP5 | 4.5 | 501 | 100 | – | – | 150 |
| GISS-E2-1-G | CMIP6 | 2.7 | 851 | 100 | 100 | – | 150 |
| GISS-E2-R | CMIP5 | 2.1 | 500 | 100 | – | 100 | 150 |
| HadGEM2-CC | CMIP5 | 4.5 | 240 | 35 | – | – | – |
| HadGEM2-ES | CMIP5 | 4.6 | 336 | 101 | – | – | 150 |
| HadGEM3-GC31-LL | CMIP6 | 5.4 | 100 | – | – | – | 150 |
| INM-CM4-8 | CMIP6 | 2.1 | 531 | 200 | 100 | 200 | 150 |
| IPSL-CM5A-LR | CMIP5 | 4.1 | 1000 | 500 | – | 200 | 150 |
| IPSL-CM6A-LR | CMIP6 | 4.5 | 1200 | 550 | 550 | – | 150 |
| MIROC-ES2L | CMIP6 | 2.7 | 500 | 100 | 100 | 100 | 150 |
| MIROC-ESM | CMIP5 | 4.7 | 630 | 100 | – | 100 | 149 |
| MPI-ESM-P | CMIP5 | 3.5 | 1156 | 100 | – | 100 | 150 |
| MPI-ESM1-2-LR | CMIP6 | 2.8 | 1000 | 500 | 100 | – | 150 |
| MRI-CGCM3 | CMIP5 | 2.6 | 500 | 100 | – | 100 | 150 |
| MRI-ESM2-0 | CMIP6 | 3.1 | 701 | 200 | – | – | 150 |
| NESM3 | CMIP6 | 3.7 | 100 | 100 | 100 | – | 150 |
| NorESM1-F | PMIP4 | 2.3 | 200 | 200 | 200 | – | – |
| NorESM2-LM | CMIP6 | 2.5 | 391 | 100 | 100 | – | 150 |
| UofT-CCSM-4 | PMIP4 | 3.2 | 100 | 100 | – | 100 | – |

the beginning and not requiring changes in the land–sea mask. The midHolocene experiment aims to replicate the conditions of 6000 years ago. The primary change in the simulation is alteration in the orbital configuration, although a small reduction in greenhouses gases is incorporated in the most recent simulations (Otto-Bliesner et al., 2017). We consider simulations from both PMIP3 (Braconnot et al., 2012) and PMIP4 (Brierley et al., 2020) together, as their forcings and boundary conditions are statistically indistinguishable.

The "lig127k" experiment represents some of the warming conditions seen during the last interglacial, focused at 127 000 years ago (Otto-Bliesner et al., 2017). The experiment is included in PMIP4 for the first time (Kageyama et al., 2018) and has been completed by 14 modelling groups (Otto-Bliesner et al., 2021). It is predominantly an orbitally forced experiment, with accompanying changes in green-house gases: the approximate +9 m of sea level rise during the last interglacial are not incorporated into the boundary conditions. The magnitude of the orbital forcing is substantially larger than seen during the midHolocene, with the anomaly in NH summer insolation being twice as strong and occurring for longer (Otto-Bliesner et al., 2017).

The world was much colder at the time of the Last Glacial Maximum (LGM), by a similar order of magnitude to the warming seen in projections (e.g. Tierney et al., 2020). As such, it has long been seen as an important test for climate models (Braconnot et al., 2012) and has featured in PMIP since its inception. The "lgm" experimental protocol (Kageyama et al., 2017) requires the imposition of large ice sheets, associated changes in the land–sea mask, and reductions in greenhouse gases. This can be a challenging experiment to deploy for a climate model, and therefore a smaller

number of modelling groups have completed it than either of the interglacial experiments (Kageyama et al., 2021). Only a handful of groups have posted output on the Earth System Grid Federation (ESGF) (Table 1) at the time of writing, so we additionally incorporate PMIP3 simulations to increase the ensemble size. Although the PMIP4 models show some differences in circulation and encompass a greater ensemble spread in temperatures, we note that Kageyama et al. (2021) conclude that they "are not fundamentally different from the PMIP3-CMIP5 results".

This research looks at coupled phenomena, and so self-consistency between all observed variables is an important factor. Therefore, we adopt a reanalysis product instead of multiple observational datasets. For this work, the 20th-Century Reanalysis (Compo et al., 2011) is used, although we do not expect our main conclusions in the model evaluation to be sensitive to this choice. Version 3 of the 20th-Century Reanalysis extends back to 1836 and uses HadISST (Rayner et al., 2003) to provide its bottom boundary conditions (Slivinski et al., 2019). Initial analysis using the shorter second version did expose a sensitivity in the IOD rainfall teleconnection over India, presumably relating to the ENSO–IOD relationship in the earlier period. Evaluation of the models is performed through comparison of the 20th-Century Reanalysis to the piControl simulations. It would be preferable to use the historical simulations instead of piControl. However, not all models have a historical simulation available, and previous work suggests the difference in biases are not substantial (Brierley and Wainer, 2018; Brown et al., 2020).

## 2.3 Analysis and definitions

The analysis undertaken in this research follows the workflow described by Zhao et al. (2022). This involves the creation of a curated replica of the relevant simulation output available on the ESGF. Then calendar adjustments are made to re-aggregate the monthly output from a present-day calendar to those representing 30° of the orbit for the past climate experiments using the PaleoCalAdjust software (Bartlein and Shafer, 2019). A modified version of the Climate Variability Diagnostics Package (CVDP; Phillips et al., 2014) is run on an individual simulation to calculate multiple pertinent time series and spatial fields. Please see Zhao et al. (2022) for further details and instructions on performing such analysis yourself.

All of the modes of climate variability explored in this research are defined through the use of area-averaged sea surface temperature anomalies (or more strictly "skin temperature", which is used as an alternative in CVDP). All anomalies are computed with respect to a climatology calculated over the full simulation length and additionally have a linear trend removed in case the simulations are not equilibrated. The same process is undertaken in the abrupt4xCO2 experiment, except only years 101–150 are considered. ENSO is monitored using the Niño3.4 region (5° S–5° N, 120–

170° W; Trenberth, 1997). The Dipole Mode Index is used to define the IOD (Saji et al., 1999). It is the difference between the area average of 10° S–10° N, 50–70° E and 0–10° S, 90–110° E. The Indian Ocean Basin Mode is defined using variations in the Tropical Indian Ocean time series (averaged over 15° S–15° N, 40–110° E; Huang et al., 2016). These modes adopt the definitions used by the IPCC (Cassou et al., 2021), who demonstrate that they are equivalent to definitions based on empirical orthogonal functions.

The El Niño-like mode of variability proposed by Thirumalai et al. (2019) is best defined using wind fields (DiNezio et al., 2020). However, those fields are not as universally available in the ESGF archive, especially for PMIP experiments. Instead, we analyse the area-averaged SST anomalies of the Eastern Equatorial Indian Ocean (EEIO) over the region 2.5° S–2.5° N, 70–95° E (DiNezio et al., 2020), the impact of which will be discussed in Sect. 4.3.

We present spatial patterns associated with the Indian Ocean Dipole, which are computed via regression. Detrended monthly anomalies of surface temperature and precipitation are computed at every grid point across the globe. These anomalies are then regressed against the Dipole Mode Index (defined as described above). The resulting patterns are therefore expressed as the local change in temperature or precipitation seen under a positive IOD event when the Dipole Mode Index is 1 °C (the level of warming reached during the extreme event in 2019; Wang and Cai, 2020). Both the IOD spatial patterns and the mean climate changes presented in Sect. 4.1 show the multi-model ensemble average. Each model's changes are computed on their own native grid and then interpolated onto a regular 1° grid before averaging across the ensemble. Stippling in figures is used to indicate where the ensemble is "not consistent" in its direction of change; this is computed as being less than two-thirds of the ensemble array on the sign of the change.

## 3 Results

### 3.1 Comparison with observations

It is necessary to assess the appropriateness of the PMIP ensemble against observations prior to looking at the response in experiments. This is performed through the inspection of the patterns of surface climate variables over the region of interest (Fig. 1). Here we use a single atmospheric reanalysis dataset (20th-Century Reanalysis; Compo et al., 2011, Sect. 2) driven by the prescribed SSTs from the HadISST dataset (Rayner et al., 2003). The annual pattern of SST consists of a pool of warm water extending from Indonesia in the east to nearly all the way across the basin (Fig. 1a). This extent is relatively well captured by the ensemble mean (Fig. 1b), although it overestimates the amount of cooler water in the Arabian Sea and near Madagascar. The annual mean precipitation pattern reflects the seasonal march of the Intertropical convergence zone (ITCZ), with an intense

oceanic band at 10° S and a more diffuse terrestrial pattern in the Northern Hemisphere (NH) especially focused over orography (Fig. 1c). The ensemble mean simulated pattern shares most of these features but is much smoother (Fig. 1d).

The Indian Ocean Dipole is characterized by a region of cooling off Java and Sumatra counterbalanced by a warming in the western half of the basin (Saji et al., 1999, Fig. 1e). This pattern in SST is fundamentally replicated by the ensemble (Fig. 1f). The models show more cooling around the Indonesian Archipelago, as well as a less extensive patch of warm SSTs in the western half of the basin, especially south of the Equator. The rainfall pattern of the IOD is strongest over the interior of the IO, with the direction controlled by the underlying SST pattern (Fig. 1g). This aspect of the rainfall pattern is well simulated by the ensemble (Fig. 1h). Both models and observations capture a reduced rainfall over India.

The IOD is positively correlated with the Indian summer monsoon rainfall (Ashok et al., 2001), although the strength of that teleconnection seems to be changing (Cherchi et al., 2021). Li et al. (2017) find that CMIP5 models (roughly half the current ensemble) have too strong a summer rainfall teleconnection. McKenna et al. (2020) show that the quality of the IOD simulation improved in CMIP6. However, this literature-based description of the IOD's teleconnection to rainfall over India is of a different sign to the response shown in Fig. 1g and h. This discrepancy results from the analysis approach used here combined with model mean state biases. Hrudya et al. (2021) demonstrate that the sign of the Indian summer monsoon IOD teleconnection is dependent on the analysis technique used (for compositing versus correlation). The IOD teleconnection shown here is computed by regressing detrended monthly anomalies (Sect. 2.3) for all seasons; this already acts to convolve IOD impacts on the summer and winter monsoons. More importantly, the regression approach we adopt can be overridden by an ENSO response which is correlated to the DMI time series (Li et al., 2017). Additionally, the models underestimate the annual rainfall over central India on average (Fig. 1c, d) because they do not capture the full strength of the summer rainfall, which further acts to reduce the summer contribution to the teleconnection. A separate investigation of the role of the Indian Ocean Dipole on the Indian monsoon rainfall using an alternate analysis approach would be required to fully isolate and understand the present and past response. It is sufficient to note here that the models do replicate the large-scale pattern emerging from the observations, but there remains doubt about the IOD teleconnections over the Indian subcontinent. Therefore, this analysis does not draw conclusions about the impact of past interannual variability changes on the Indian monsoon.

## 3.2 Changes in mean climate

The tropical climate in the experiments analysed here primarily responds to one of two forcings: changes in the orbital

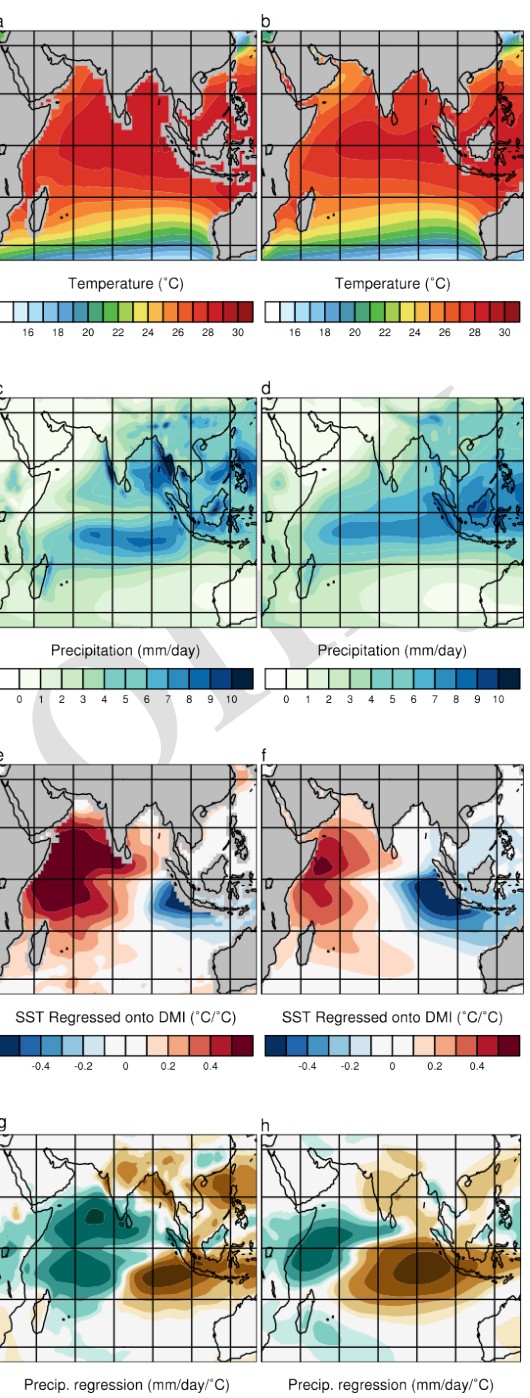

**Figure 1.** Comparison between the observed climate and the PMIP ensemble. The left-hand side is taken from the 20th-Century Reanalysis (Slivinski et al., 2019), with the ensemble mean of the preindustrial control simulations on the right. The rows present the annual mean sea surface temperatures **(a, b)**, annual mean precipitation **(c, d)**, IOD pattern computed by regressing the monthly SST anomalies against the DMI **(e, f**; Sect. 2.3), and the rainfall anomalies associated with IOD variations **(g, h)**.

configuration or in the concentrations of greenhouse gases. These different forcings result in two distinct climate responses, as demonstrated in the simulations. First we present and discuss the midHolocene and lig127k experiments together and then move onto a description of the lgm and abrupt4xCO2 experiments. We note this latter comparison is not perfect because the high-latitude ice sheets do impact the tropics through bathymetry changes (DiNezio and Tierney, 2013) and atmospheric teleconnections (Lee et al., 2015; Lu et al., 2016; Jones et al., 2018; Ullman et al., 2014).

Precession leads to changes in the day of the year at which the Earth is closest to the Sun (the perihelion). At present, this occurs during NH winter, yet this was nearer the autumn equinox during the mid-Holocene and occurred near the summer solstice at 127 ka (Berger and Loutre, 1991). This results in a different distribution of the incoming solar radiation throughout the year. The eccentricity of the orbit was also larger during the last interglacial, exacerbating these changes in the lig127k experiment (Otto-Bliesner et al., 2017). Intuitively, more sunlight in summer and less in winter will lead to a stronger seasonal temperature range in the Northern Hemisphere, especially outside of the tropics. This is visible in Fig. 2b, c for the midHolocene simulations (Brierley et al., 2020) in Asia north of the Himalayas and even more so in lig127k (Fig. 2e, f; Otto-Bliesner et al., 2021). Over most of the region, the seasonal temperature changes "cancel" themselves out, leading to minimal changes in the annual mean (Fig. 2a, d). This is especially true over the ocean, where its high effective heat capacity acts to damp seasonal variations.

There is an obvious exception to the simple narrative of precession-modulated temperature response outlined above, with summer cooling over land stretching from the Indian subcontinent over Arabia and into Ethiopia (Fig. 2c, f). This can be thought of as being related to a poleward shift in the summer ITCZ driven by the stronger interhemispheric temperature gradients (Braconnot et al., 2007; Yeung et al., 2021), mainly through dynamic processes rather than thermodynamic ones (D'Agostino et al., 2019). The impact of this process is greatest over West Africa and is underestimated in model simulations (Perez-Sanz et al., 2014; Brierley et al., 2020), and both experiments fall under separate instances of an African humid period (Ziegler et al., 2010). Summer rainfall increases over East Africa are clearly visible (Fig. 3c, f), which extend into the Arabian peninsula. This increased rainfall comes with an increase in convection and cloud cover, leading to a reduction in incoming solar radiation at the surface as well as latent heat flux changes associated with the precipitation. The situation is different over India, however. There is no increase in total rainfall in the Indian summer monsoon (Fig. 3, Brierley et al., 2020) but rather a redistribution of it, so that more falls in the foothills of the Himalayas instead of on the central Indo-Gangetic Plain. A similar style of dipole response occurs over the Western Ghats. The band of anomalous summer drying extends east from central India over South-East Asia to the

Philippines (Brierley et al., 2020; Otto-Bliesner et al., 2021). Meanwhile the Indonesian Archipelago is wetter in JJA and drier in DJF, as is northern Australia.

Over the Indian Ocean itself, there is an orbitally driven increase in DJF rainfall that is strongest near the African coast (Fig. 3b, e). This accompanies, and could be driving, a slight cooling in DJF in the region (Fig. 2b, e). In JJA, the orbital forcing results in a west–east dipole of rainfall changes (Fig. 3c, f), which is reminiscent of the IOD teleconnection (Fig. 1h). This is accompanied by some low-amplitude, annual mean SST changes that project onto the IOD (Fig. 1h), although these are only visible in the lig127k ensemble (Fig. 2d). The weaker orbital forcing in the mid-Holocene experiment means that the changes are swamped by the impact of the reduced greenhouse gases in the PMIP4 experiment (Brierley et al., 2020).

The Last Glacial Maximum was a substantially colder period, with global mean cooling of around 5–7 °C (Gulev et al., 2021). The lower greenhouse gas concentrations (Kageyama et al., 2017) played a dominant role in determining the tropical climate changes, although impacts on climate variability from the Laurentide ice sheet were felt across the globe (Jones et al., 2018). The tropical Indian Ocean itself cools by 2–3 °C in the PMIP ensemble (Fig. 2g), roughly in line with estimates from palaeodata assimilation (Tierney et al., 2020). Conversely a quadrupling of $CO_2$ leads to an increase in temperatures by 4–5 °C (Fig. 2j). Both responses show little seasonal variation and greater amplitudes of temperature change over the land (Fig. 2). The two patterns mirror each other to first order, except for a stronger response over the Indonesian Archipelago associated with the exposure of the Sunda Shelf (DiNezio and Tierney, 2013).

The colder climate of the lgm experiment leads to drier conditions across the region (Fig. 3g; DiNezio and Tierney, 2013), whilst future warming leads to increased rainfall across most of the basin (Fig. 3j). There is greater seasonal variation in the rainfall changes (Fig. 3k, l) than the temperature response (Fig. 2k, l), reflecting the stronger seasonal cycle in rainfall. There is an interesting non-linearity in the JJA rainfall changes between the LGM and idealized warming to the south-east of Sumatra and Java (cf. Figs. 3g and j) in that both high- and low-$CO_2$ experiments demonstrate a drying in the region. This non-linearity is centred on the eastern region of the Dipole Mode Index (Sect. 2.3) and therefore could be expected to influence the response of the Indian Ocean Dipole. This arises from the exposure of the Sunda Shelf (DiNezio and Tierney, 2013; DiNezio et al., 2018), which has already been shown to have an influence on the ensuing variability (Thirumalai et al., 2019). Even though the underlying drivers of climatic forcing for the lgm and abrupt4xco2 experiments differ, they trigger a similar mean zonal response in the tropical Indian Ocean that has been suggested to modulate interannual variability across the basin (Thirumalai et al., 2019; DiNezio et al., 2020).

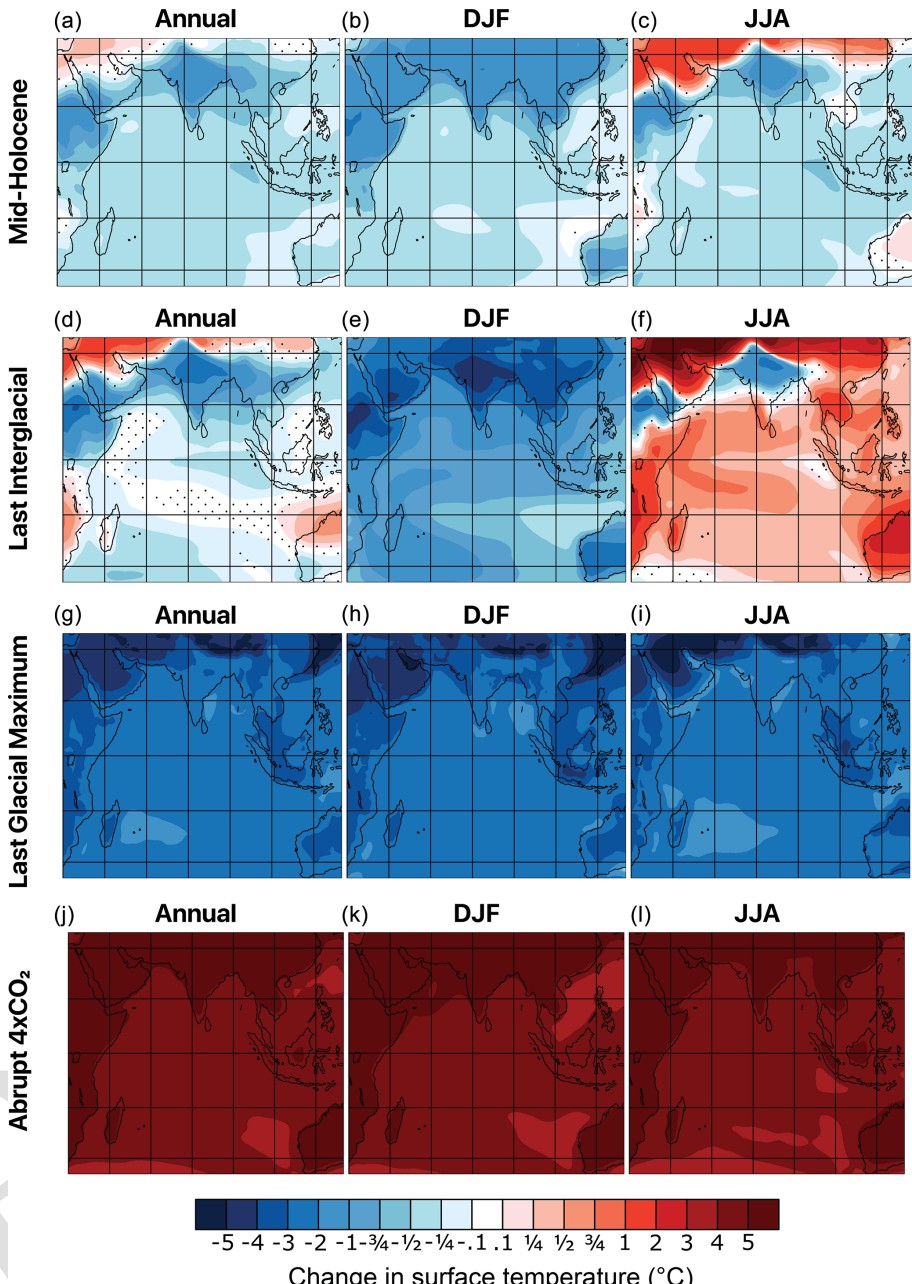

**Figure 2.** Ensemble mean change in surface temperatures. The columns show the annual mean **(a, d, g, j)**, DJF **(b, e, h, k)**, and JJA **(c, f, i, l)**. The rows show the ensemble mean difference from the piControl simulations for the midHolocene **(a, b, c)**, lig127k **(d, e, f)**, LGM **(g, h, i)**, and abrupt4xCO2 simulations **(j, k, l)**. The ensemble mean of the respective piControl simulations is overlaid as black contours (shown for every $2\,\mathrm{mm\,d^{-1}}$ until $12\,\mathrm{mm\,d^{-1}}$). Stippling indicates where the ensemble is not consistent in the direction of change.

## 3.3   Changes in variability

Having established that the experiments lead to consistent changes in the mean Indian Ocean climate, we next explore whether they also alter its variability. We start by looking at the IOD, whose pattern is relatively well captured by the models (Sect. 3.1). The amplitude of the IOD can be characterized by the standard deviation of the monthly Dipole Mode Index (Sect. 2.3). The standard deviation of the (detrended) Dipole Mode Index in the HadISST dataset, used by the 20th-Century Reanalysis, is $0.46\,°C$. There is a spread in the amplitude of the IOD in the piControl simulations that ranges from $0.24\,°C$ in INM-CM4-8 to $0.75\,°C$ in CSIRO-Mk3-6-0, with an ensemble median that is close to the observed value (Fig. 4a). The precise composition of models that have undertaken each experiment varies, leading to subtly differ-

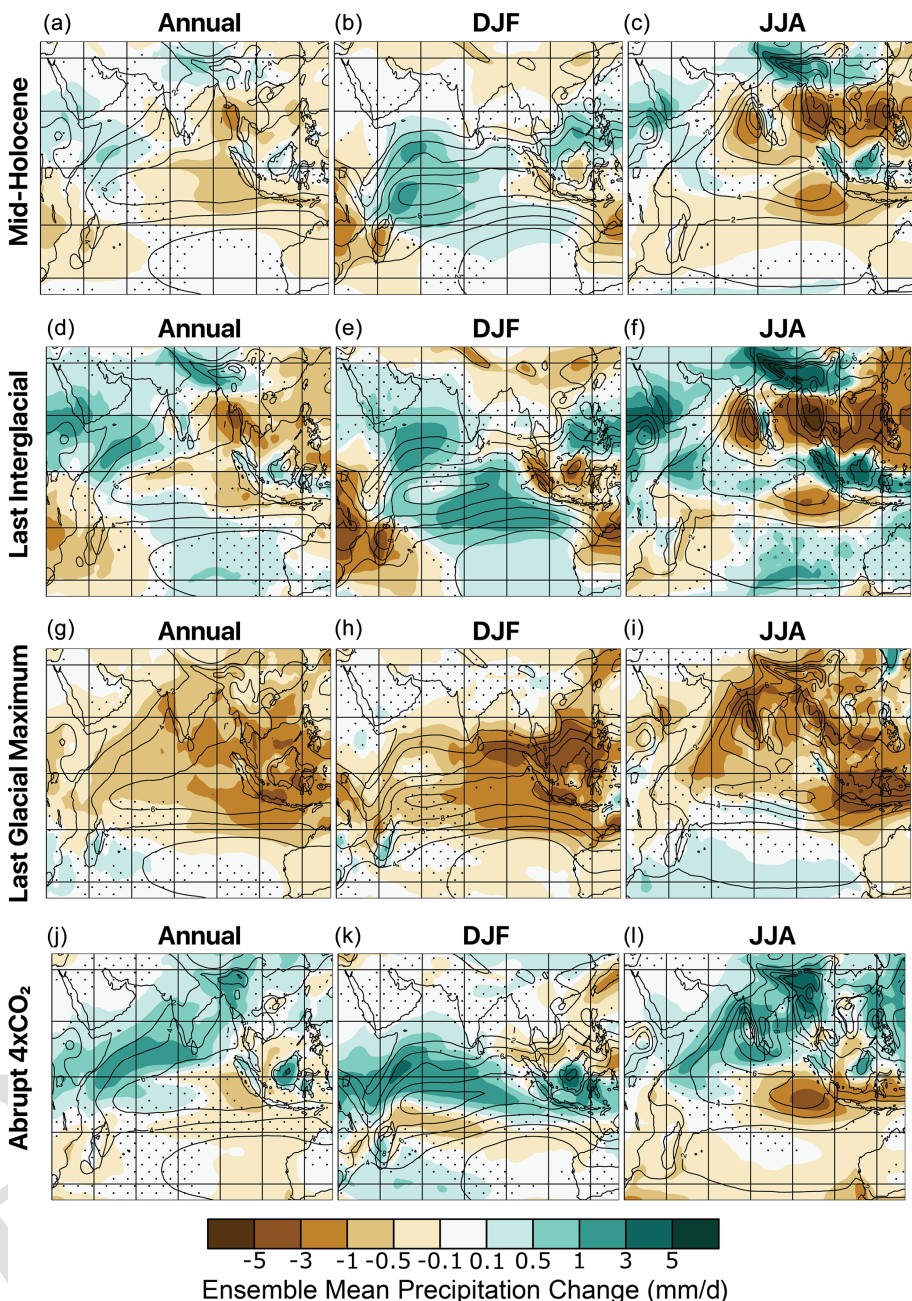

**Figure 3.** Ensemble mean change in precipitation. The columns show the annual mean (**a, d, g, j**), DJF (**b, e, h, k**), and JJA (**c, f, i, l**). The rows show the ensemble mean difference from the piControl simulations for the midHolocene (**a, b, c**), lig127k (**d, e, f**), LGM (**g, h, i**), and abrupt4xCO2 simulations (**j, k, l**). Stippling indicates where the ensemble is not consistent in the direction of change.

ent control ensembles. Those models that have performed an LGM experiment are generally ambiguous about its impact on the IOD amplitude: 7 of the 13 models show a slight decrease, with the other 6 showing an increase. MIROC-ESM shows the largest change (an increase of 63 %), although its successor model has a very small reduction ($-0.01\,^{\circ}$C). The median value also remains the same (Fig. 4a), leading us to conclude to that the last glacial conditions did not im-

pact the magnitude of the Indian Ocean Dipole. A similar situation occurs for the lig127k experiment: equivocal simulated changes leading to minimal changes in the IOD amplitude distribution. Despite having a weaker orbital forcing, the midHolocene sees a reduction in the standard deviation of the DMI of roughly 10 %. The individual models within the ensemble still show some disagreement about the signal, with only 20 out of 29 models simulating a re-

duction of some magnitude (CSIRO-Mk3-6-0 is an outlier, whose already high IOD amplitude increases by a further 35 %). There is a robust decrease in IOD amplitude in the abrupt4xCO2 experiments (Fig. 4; taken between years 101-150 with a linear trend removed), with 18 of the 24 simulations showing a reduction of some magnitude. This result was noted in PMIP3 by Rehfeld et al. (2020) and fits with Cai et al. (2013), who anticipate a reduction in amplitude should the mean state changes resemble a positive IOD, conditions that are found in Fig. 2l, although this will be explored in more in Sect. 4.1. The abrupt4xCO2 experiments are still not equilibrated, and there may be some impacts of slow feedbacks that have not yet fully emerged (Heede et al., 2021).

Our approach to define the IOD pattern through linear regression (Sect. 2.3) allows spatial changes to be considered separately from changes in amplitude. The orbital forcing experiments both see an expansion of the IOD cold pole eastwards along the Equator (Fig. 5a, c), which is stronger in the lig127k than the midHolocene. This spills out over the Equator into the Northern Hemisphere, especially into the Bay of Bengal. In the lig127k experiment, this acts to make Bay of Bengal SSTs cool during a positive IOD event in every model. This is a region where many models already have trouble replicating the warming signal seen in observations (Fig. 1e, f), but even the five models that successfully capture this in the piControl simulations switch to cold SSTs in the lig127k (not shown). This is accompanied by a constriction of the western SST pole of the IOD pattern back towards the East African coast (Fig. 5c). This results in the boundary between the dipolar phases of the IOD occurring to the west of Sri Lanka, rather than to its east, in the majority of models. The shifts in the midHolocene experiments are not as strong but are of a similar nature (Fig. 5a, c).

The expansion of the SST cold pole along the Equator during the orbital simulations (Fig. 5a, c) is accompanied by a reduction in rainfall that extends further westward during a positive IOD (Fig. 5b, d). There is also a slight weakening of the strong drying response in the south-east Indian Ocean (east of Java) and an intensification of the wet conditions off the East African coast, which do not extend into the continent. Both the lig127k and midHolocene ensembles suggest that the IOD had a wetter response over central India (Fig. 5b, d), although this could indicate a greater contribution from the IOD rather than ENSO throughout the simulation (see Sect. 3.1).

There is little change in the SST pattern of the IOD in the LGM experiment (Fig. 5c), with the only substantial changes occurring within the Maritime Continent. This is likely associated with the sub-aerial exposure of the Sunda Shelf (Kageyama et al., 2017), which is handled differently by the different models and with varied responses in the atmospheric circulation (DiNezio and Tierney, 2013). The rainfall teleconnection pattern associated with the IOD is severely damped in the LGM experiment. This likely occurs as a consequence of the Clausius–Clapeyron relationship; the regres-

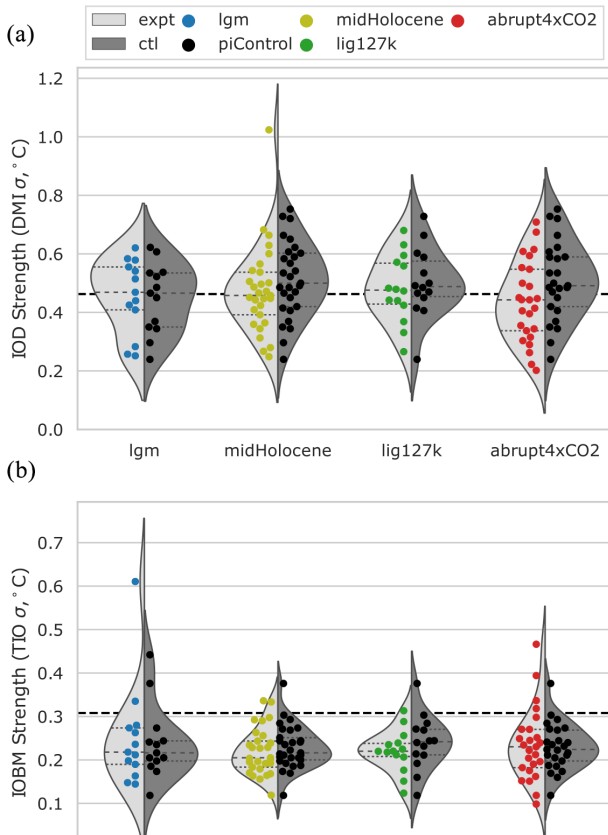

**Figure 4.** Amplitude changes in modes of Indian Ocean variability. **(a)** The amplitude of the Indian Ocean Dipole (IOD) is measured by the standard deviation of the Dipole Mode Index (DMI; the difference in area-averaged SST anomalies between the western and south-eastern Indian Ocean). **(b)** The amplitude of the Indian Ocean Basin Mode (IOBM) is measured by the standard deviation of the tropical Indian Ocean Index. See Sect. 2.3 for definitions of the areas over which the SST anomalies are computed. The values computed using the 20th-Century Reanalysis are shown as black dashed lines. The distributions are computed using a kernel density estimation (Waskom, 2021) from the simulation values shown, as are the quartiles shown by the short dashed and dotted lines. The horizontal location of the dots has no meaning and has been chosen to allow all simulations to be seen.

sion is performed on absolute rather than relative rainfall, and there is a substantial reduction in climatological precipitation across the tropics (Fig. 3g).

The change in the IOD pattern of SST in the abrupt4xCO2 experiment (Fig. 5d) sees an equatorward shift in the cold pole, combined with a slight expansion westwards along the Equator. The ensemble does not show a consistent response of changes in the warm pole, although there is some consistency in the Arabian Sea with most models showing an expansion of the warm pole there. Interestingly, in the abrupt4xCO2 experiment anomalous rainfall variations as-

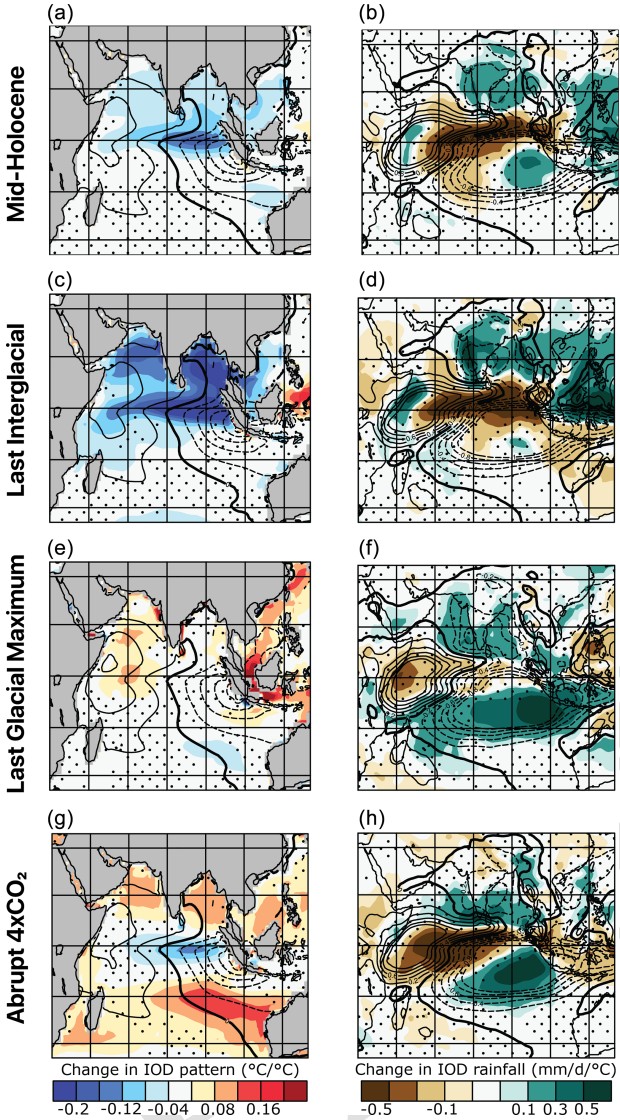

**Figure 5.** Ensemble mean change in SST and rainfall patterns of the Indian Ocean Dipole. Each row represents the ensemble mean difference from the piControl simulations and a different experiment, with the midHolocene changes in SST **(a)** and rainfall **(b)**, the lig127k changes in SST **(c)** and rainfall **(d)**, the LGM changes in SST **(e)** and rainfall **(f)**, and the abrupt4xCO2 changes in SST **(g)** and rainfall **(h)** patterns. The patterns within each simulation are first computed through regressing monthly anomalies onto the Dipole Mode Index; then each model's changes in those patterns are averaged across the ensemble (see Sect. 2.3). Stippling indicates where the ensemble is not consistent in the direction of change (i.e. less than two-thirds agree on it). The overlaid contours show the ensemble mean pattern in the piControl simulations, with dashed contours indicating negative numbers. As the models contributing to each experiment change, so does the precise pattern of piControl patterns.

sociated with positive IOD events (which generate drying over the east – see Fig. 1h) reduce across the Indian Ocean (Fig. 5h).

The Indian Ocean Basin Mode (IOBM) involves synchronous fluctuations across the whole basin. Its amplitude can be measured by looking at the standard deviation of the tropical Indian Ocean (Fig. 4b). The PMIP4 ensemble shows large variations in the quality of their simulation in the piControl, ranging from roughly one-third to 1.5 times the amplitude seen in the 20th-Century Reanalysis (0.31 °C Compo et al., 2011). In the LGM experiment, there is no change in the median IOBM amplitude, despite more than two-thirds of the ensemble suggesting an increase and COSMOS-ASO simulating a dramatic one. The midHolocene ensemble shows a slight reduction in IOBM amplitude, seen in 22 of the 28 members with a mean reduction of −3.7 %. A similar behaviour is seen in the lig127k ensemble, although the amplitude reduction is slightly stronger, with an ensemble mean reduction of −6.4 % (Fig. 4b). There is a slight increase in IOBM amplitude under the abrupt4xCO2 experiment (as suggested by, e.g., Tao et al., 2015), although the ensemble as a whole is equivocal, with only 12 of the 25 members agreeing in sign with this mean.

## 4 Discussion

### 4.1 Relationship with the mean state

In this work, we have looked at both the response of the mean state of the Indian Ocean and its variability under a variety of experiments. We found changes in both of these properties and have discussed some spatial relationships between their ensemble mean responses. A natural extension of this is to ask if there exist underlying relationships that act across all the different experiments. The changes in the Indian Ocean Basin Mode (Fig. 4b) were weak in every experiment, yet there were substantial basin-wide SST changes in the LGM and abrupt4xCO2 experiment (Fig. 2). It should therefore come as little surprise that PMIP4–CMIP6 demonstrates no significant relationship between the changes in the mean tropical Indian Ocean Index and changes in its standard deviation (Fig. 6a).

The literature suggests that changes in the mean state of the Indian Ocean can map onto the Dipole Mode Index (Cai et al., 2013). This is further suggested to result in changes in the IOD (Cai et al., 2013) via changes in the equatorial thermocline (Zheng et al., 2013). The PMIP4–CMIP6 ensemble shows little consistency among changes in the mean gradient between the two poles of the Dipole Mode Index and the variability of the index (Fig. 6b). Even separating out just individual experiments shows a muted relationship. Given the large ensemble size and that the majority of the experiments have reached quasi-equilibrium, there is little scope for the initial conditions of internal variability to influence the IOD changes under this experiment, highlighted as an important

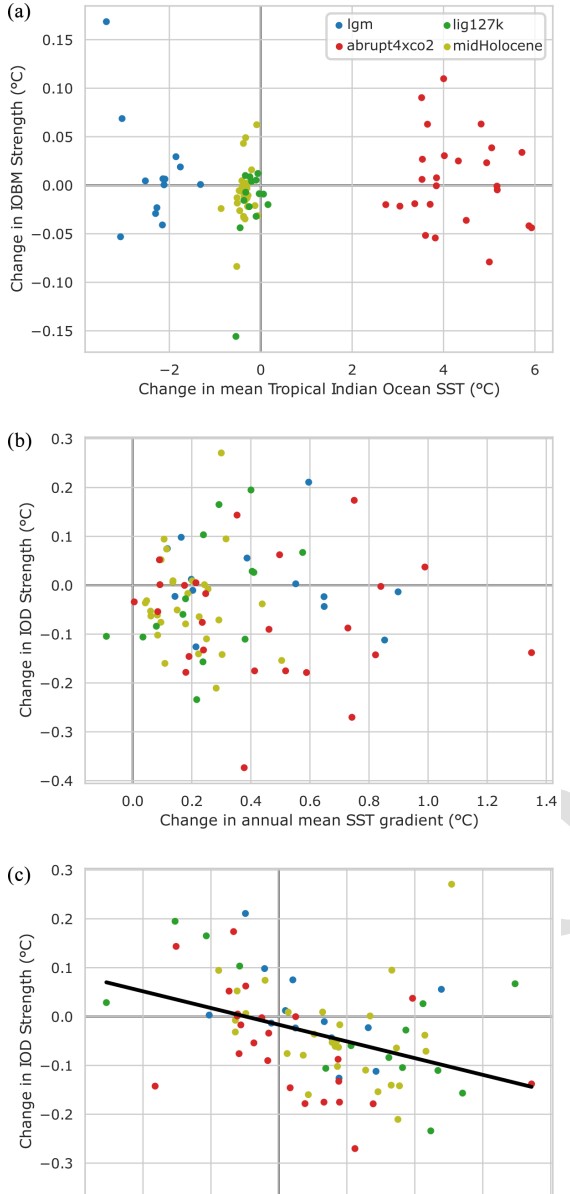

**Figure 6.** Relationships between the amplitude changes in modes of Indian Ocean variability and the mean state. **(a)** The strength of the Indian Ocean Basin Mode does not respond to changes in the tropical Indian Ocean SST. **(b)** The strength of the Indian Ocean Dipole mode does not exhibit a simple linear relationship with annual mean changes in the zonal gradient. **(c)** Changes in IOD strength are connected with changes in seasonal cycle of the SST gradient, here represented as changes in the differences between the strongest and weakest monthly SST gradients. The black solid line indicates a statistically significant linear regression across the whole ensemble.

source of uncertainty in future projections by Hui and Zheng (2018). It should be noted that the method adopted here not only combines positive and negative modes of the IOD but also moderate and strong positive events, which have been shown to respond differently under future warming (Cai et al., 2021b). However, orbital changes result in different seasonal temperature changes in the west and east Indian Ocean, most visibly in the lig127k experiment (Fig. 2e, f). This means that the annual cycle of the SST difference measured by the Dipole Mode Index varies under orbital forcing (Brown et al., 2009; Iwakiri and Watanabe, 2019) and also happens to occur in the other experiments. One way to explore this seasonality is to look the amplitude of the annual cycle (i.e. the difference between the maximum and minimum monthly SST gradients). Changes in the amplitude of the seasonal cycle are inversely proportional to the changes in the IOD strength (Fig. 6c). Further independent research supports the importance of the changes in seasonal cycle in the midHolocene ensemble (Liu et al., 2023) and looks at the physical mechanisms behind their influence. Taken together these findings suggest there is potential to constrain the future projections of the IOD with (palaeo-) observations of the seasonal change in SST gradients.

## 4.2   Pacific influence via ENSO

There is a strong link between variability in the Pacific Ocean, predominantly the El Niño–Southern Oscillation (ENSO), and that in the Indian Ocean. This impacts both the IOD (Stuecker et al., 2017) and the IOBM (Xu et al., 2021). Brown et al. (2020) have already explored changes in ENSO across the PMIP4–CMIP6 ensemble. They saw consistent reductions in ENSO amplitude in both the lig127k and midHolocene experiments but little consistency across the other two experiments presented here. Similar to the results shown in Fig. 6a and b, there is no universal relationship between mean state changes in the equatorial Pacific and the amplitude of ENSO (Brown et al., 2020). The amplitude of ENSO in the various experiments, as measured by the standard deviation of the smoothed monthly Niño3.4 SST anomalies, is shown in Fig. 7a. The damping of ENSO in the orbital simulations is most visible in the lig127k experiment. There are hints of an increase in the abrupt4xCO2 experiment; however, the ensemble is equivocal, with 11 out of 25 models showing a reduction in ENSO amplitude. Despite this lack of consistency in the simulated direction of change, the ensemble demonstrates a strong positive relationship between the changes in the amplitude of ENSO and those of the IOBM (Fig. 7b), which is statistically significant. This conforms nicely with our pre-existing knowledge of the role of ENSO as a driver for IOBM events (Xu et al., 2021).

The relationship between changes in ENSO and the IOD are more nuanced. ENSO is known to play a role in a proportion of IOD events (Ashok et al., 2003), although by no means all of them (Kajtar et al., 2017). Therefore if ENSO is

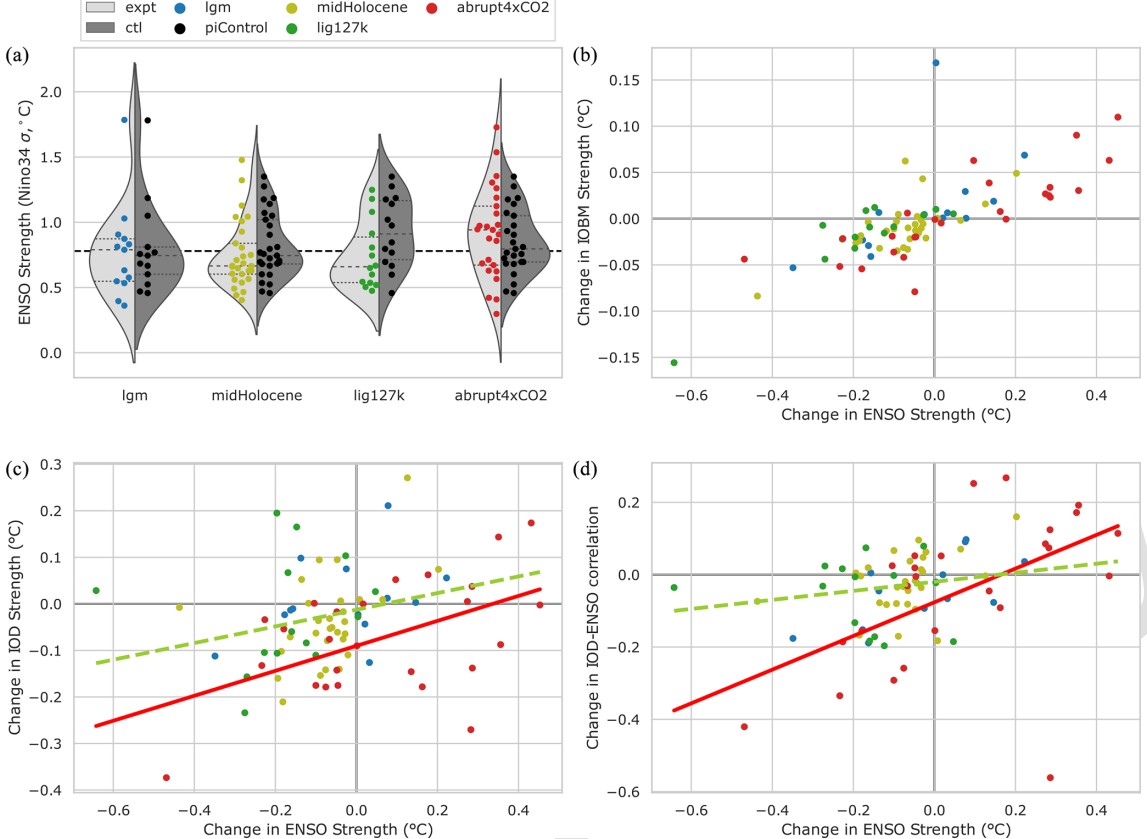

**Figure 7.** Relationships between the amplitude changes in modes of Indian Ocean variability and the mean state. **(a)** The amplitude of Pacific ENSO is measured by the standard deviation of the Niño3.4 SST anomalies (Sect. 2.3). **(b)** Amplitude changes in the IOBM are strongly related to changes in ENSO amplitude. **(c)** Amplitude changes in the IOD show some signs of being related of changes in ENSO amplitude. **(d)** There is a weak indication suggesting that as ENSO amplitude becomes larger, its correlation with the IOD also strengthens. The red solid lines show a statistically significant linear relationship in the abrupt4xCO2 experiment. Yellow–green dashed lines show the linear regressions across the two orbital experiments that are not statistically significant.

damped one might reasonably expect that the IOD also becomes weaker. Alternatively one could envisage that ENSO's relative importance for the IOD becomes less, as has been posited before for palaeoclimate periods (Thirumalai et al., 2019; Liu et al., 2007). We explore both possibilities through scatter plots between and the ENSO amplitude (Fig. 7c) and the IOD amplitude (Fig. 7d). Statistically significant lines of best fit could be drawn in both cases for the ensemble as a whole, yet it is unclear if this a wise choice. There is a strong contribution from the abrupt4xco2 experiments between changes in the amplitude of ENSO and both the IOD amplitude and its correlation with ENSO (shown as solid red lines in Fig. 7c, d). However, neither line passes through the origin, which would be required as these are scatter plots of anomalies from the piControl and thus call for invoking model biases or other uncertainties. Equally, the relationships are not supported across the pair of orbital experiments, both of which do show a linear regression of the same sign (yellow–green lines in Fig. 7c, d) but neither of which are statistically significant.

### 4.3 Niño-like mode

Thirumalai et al. (2019) propose the existence of a third mode of interannual variability in the Indian Ocean, unobserved in the modern climate but which may have been active during the Last Glacial Maximum. Sediment cores in the eastern equatorial basin display evidence of interannual variability that cannot be explained by either the IOD or the IOBM alone. Instead, Thirumalai et al. (2019) demonstrate that an equatorial mode, characteristically similar to ENSO, drove SST variability in much the same way as is currently observed in the equatorial Pacific. The disappearance and possible future re-emergence of this mode are related to changes in the mean state tropical climate (DiNezio et al., 2020). Few PMIP models are capable of capturing the changes in regional hydroclimate inferred from LGM proxies (DiNezio and Tierney, 2013). However, those that can, also exhibit a weakening of the Indian Walker circulation induced by the exposure of the Sunda and Sahul shelves. Anomalous divergence and easterly wind anomalies over the

exposed shelves then increase coastal upwelling off Indonesia, shoaling the thermocline and steepening the equatorial SST gradient (Di Nezio et al., 2016). These conditions favour a mode of variability which, unlike the IOD, is characterized by a distinctively equatorial pattern with interannual warm and cool events of equivalent magnitude (Thirumalai et al., 2019). Similar to a Pacific El Niño, its warm phase is preceded by a Kelvin wave in the thermocline that propagates from west to east. The projected response of the IO to greenhouse warming bears similarities to the simulated LGM, including a shoaling of the thermocline and a steeper equatorial SST gradient (Zheng et al., 2013). It has therefore been proposed that this Indian Ocean "Niño-like mode" could re-emerge in certain emission scenarios (DiNezio et al., 2020).

Here, we define the Niño-like mode index as the standard deviation of SST anomalies in the eastern equatorial Indian Ocean (EEIO; Sect. 2.3). DiNezio et al. (2020) demonstrate that SST anomalies are at best an approximation for the Niño-like mode, since the equatorial shift in the IOD in some climates can also contribute to SST anomalies in this region. Ideally, the mode would be identified by its atmospheric precursor: wind anomalies in the western basin. However, these fields are not universally available. The EEIO region overlaps in its bottom corner with that used to identify the southern-eastern pole of the IOD and is clearly contained in the basin-wide anomalies used to track the IOBM. Nonetheless, we note that the IOD influence over equatorial SSTs is small relative to the Niño-like mode in simulations by DiNezio et al. (2020), and there is little relationship between the amplitude changes in IOD and EEIO (not shown). The amplitude of the Niño-like mode in each experiment is shown in Fig. 8a. The piControl ensemble exhibits a range of standard deviations between 0.18 and 0.53 °C, yet the ensemble median (0.37 °C) is rather close to that of the reanalysis (0.38 °C).

The most robust response is observed in the lig127k ensemble, in which 13 of 14 models agree on an increase in amplitude during the last interglacial. Only MIROC-ES2L simulates a decrease in variability (of 46 %) with the remaining models averaging a 17 % increase. In contrast, the other experiments show little to no agreement between models, with the mean amplitude going largely unchanged from the piControl. Even the lgm experiment, which is the only past climatic period previously analysed for the existence of this mode, has little agreement: the amplitude increases in six models and decreases in six models (the substantial increase from COSMOS-ASO does shift the ensemble mean in Fig. 8a though). Across PMIP, a roughly 1 : 1 relationship emerges between amplitude changes in variability within the EEIO region and across the whole basin (as measured by IOBM strength), which is statistically significant. This behaviour would be expected in the absence of the Niño-like mode. However, the relationship does not exist for the lig127k ensemble, where the consistent increase in EEIO standard deviation (Fig. 8a) occurs despite a decrease in variability across the whole basin (Fig. 4b). It is also interesting that the dom-

inant period of interannual variability of in the EEIO region reduces under orbital forcing (not shown; see the "Code and data availability" section). These two observations provide further evidence that a Niño-like mode can emerge under certain climate states.

Thirumalai et al. (2019) and DiNezio et al. (2020) both use CESM1.2 in their model analysis. However, its CMIP6 successor, CESM2, lacks an lgm experiment at the time of writing. Its predecessor, CCSM4, is included here and exhibits an 8 % decline in variability. However, Di Nezio et al. (2016) explicitly note that CCSM4 fails to simulate the eastern drying and western wetting reconstructed from proxies, as do most CMIP5 models. They argue that this is due to its inadequate atmospheric response to shelf exposure, a key mechanism in the onset of the Niño-like mode. Although this is the first analysis of the Niño-like mode in CMIP6, we acknowledge that this systematic bias may persist into our results. An updated review of PMIP4 with respect to glacial hydroclimate proxies, similar to DiNezio and Tierney (2013), would be required to clarify this point. Nonetheless, the robust increase in variability in the lig127k experiment presents an opportunity for the Niño-like mode to be studied in past climates other than the LGM.

How the Niño-like mode interacts with other modes of variability is relatively unexplored. The proximity of the Niño-like mode's equatorial arm to the IOD region off of Sumatra and Java and the observation that both modes peak in boreal autumn lead to the reasonable assumption that the two might influence each another through either changes in mean state or variability. DiNezio et al. (2020) link the future re-emergence of the Niño mode with a shoaling of the thermocline and enhanced upwelling in the eastern basin, conditions which resemble a positive IOD. Furthermore, there is a realistic possibility that ENSO could influence the Niño-like mode as it does the IOBM (Fig. 7b) and, to a lesser extent, the IOD (Fig. 7c). Figure 8b shows a statistically significant relationship between the mean equatorial SST gradient and the amplitude of the Niño-like mode in the lig127k experiment, which also passes through the origin. However, this relationship is not shared across the rest of the ensemble. Potentially this arises from the strong and consistent changes in the seasonality seen in the lig127k experiment, as this was shown to impact the IOD (Fig. 6c).

We also investigate influences on the relationship between the Niño-like mode and the IOD through an analysis of their correlation. Although Thirumalai et al. (2019) demonstrated the independence of the Niño-like mode through its unique precursors, neither they nor DiNezio et al. (2020) quantified its synchronicity with the IOD. Here, we find that 88 of 114 simulations across all experiments exhibit a negative correlation between the Dipole Mode Index and the EEIO index, with a mean of correlation coefficient of −0.31. The strong positive IOD event in 1997–1998 was associated with ENSO and produced SST anomalies in the EEIO (Huang et al., 2022). However, there is no connection between changes in

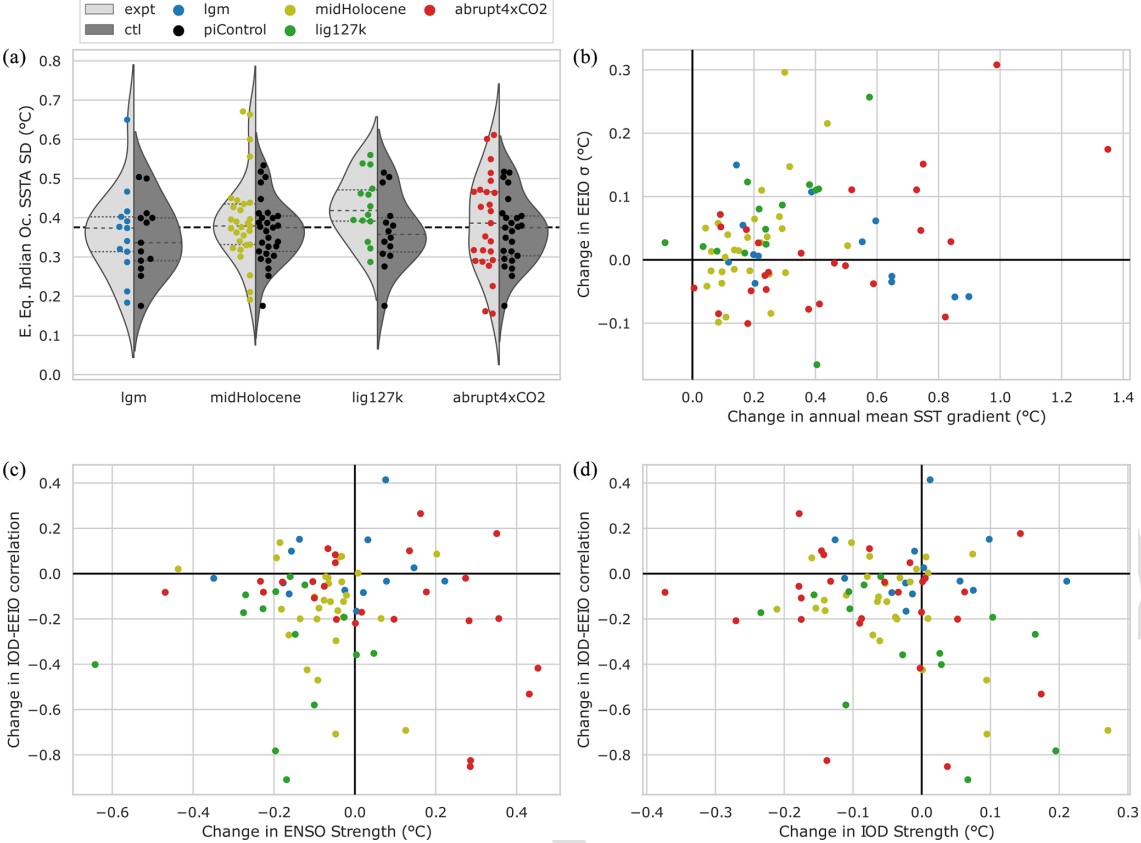

**Figure 8.** Response of Niño modes. **(a)** The amplitude of the Indian Ocean Niño-like mode is measured by the standard deviation of the eastern equatorial Indian Ocean SST anomalies (see Sect. 2.3 for definition). **(b)** The relationship between variability changes in the eastern equatorial Indian Ocean and the mean zonal gradient across the Indian Ocean, measured by the DMI. **(c)** The changes in the correlation coefficient between the two Niño modes, as a function of changes in the amplitude of Pacific ENSO. **(d)** The changes in the correlation coefficient between the Indian Ocean Dipole and the Indian Ocean Niño-like mode, as a function of changes in the amplitude of the Indian Ocean Dipole.

the correlation of the IOD and Niño-like modes and changes in either ENSO (Fig. 8c) or the IOD amplitude (Fig. 8d). We also find that the correlation between the DMI and EEIO index becomes stronger (i.e. more negative) in all models during the lig127k experiment. Distinguishing between a "switching on" of the Niño-like mode in the lig127k experiment and the IOD pattern bleeding into equatorial regions (Fig. 5c, similar to the strong positive IOD in 1997–1998) is challenging. Further detailed investigation into the role of changes in seasonality and phase locking of both modes and their atmospheric precursor would be particularly useful.

## 5  Conclusions

Using PMIP4 simulations, we have explored how changes in mean climatic Indian Ocean conditions affect its multiple modes of interannual variability across both globally warm and cool palaeoclimate. From this model ensemble, we focus on four palaeoclimate intervals that allow us to assess the response to orbital ("midHolocene", "lig127k") and green-

house gas forcing ("lgm", "abrupt4xCO2"). We find ensemble mean climatic differences between these experiments and ones forced under preindustrial conditions to be in line with expectations. Yet, across all simulations, there was no systematic relationship between indices of coupled climate variability and zonal SST gradients. Models reproduce observed patterns of IO variability but show different responses to the disparate forcing scenarios. No robust changes are seen in any interannual mode, including the IOD, IOBM, and a recently hypothesized interannual mode of equatorial variability (the Niño-like mode) for the greenhouse gas experiments (lgm and abrupt4xCO2). Under orbital forcing, the IOD robustly weakens in the midHolocene experiment but not in lig127k, which shows a small ensemble mean weakening. Both experiments suggest that the altered orbit results in an IOD with an extended cold pole along the Equator, linked to anomalous rainfall responses over the central Indian Ocean. Interestingly, orbital damping of ENSO also damps the IOBM, a result robust across both experiments. Characterized by SST anomalies in the eastern equatorial In-

dian Ocean (EEIO), the Niño-like mode shows a robust increase in the lig127k experiment and is slightly increased under the lgm experiment. Only the coupling of the changes in the strength of the Indian Ocean Basin Mode (IOBM) with changes in ENSO strength was consistent across experiments: neither the IOD nor Niño-like mode showed robust relationships in any experiments. We find that proposed relationships between changes in the mean zonal SST gradient and IOD strength are not universal, although we find suggestions of a more nuanced relationship including variations in the seasonal cycle. Further work is required to fully understand the role of seasonality shifts in IOD changes, and reconstructions of their past changes have the potential to constrain model projections of future changes in Indian Ocean variability.

**Code and data availability.** NOAA/CIRES/DOE 20th-Century Reanalysis (V3) data provided by the NOAA PSL, Boulder, Colorado, USA, from their website at https://psl.noaa.gov/data/gridded/data.20thC_ReanV3.html (Slivinski et al., 2023). Climate simulation output is freely available to download from the Earth System Grid Federation. Outputs from PMIP3 and PMIP4 have been curated and processed following the workflow outlined by Zhao et al. (2022) and are available from them.

The codes used create the figures, along with data actually plotted, are available at https://github.com/pmip4/IndianOceanVariability (last access: 21 February 2023; https://doi.org/10.5281/zenodo.7636502, Brierley, 2023). This includes a netcdf file for every simulation containing the mean climate states, the IOD pattern, and time series of every climate mode discussed. There are also ensemble mean fields computed on the regular grid used by the IPCC's Interactive Atlas. The repository also has a single spreadsheet (summary_data/tidy_numbers.csv) compiling key statistics for all the simulations, such as the amplitude and dominant period for each of the four modes investigated here.

**Author contributions.** CB and KT jointly conceived and coordinated the research. CB wrote the code and most of the first draft of the paper. EG performed initial analysis and wrote Sect. 1. JB was responsible for Sect. 4.3 on the Indian Ocean Niño-like mode. KT edited the whole document.

**Competing interests.** The contact author has declared that none of the authors has any competing interests.

**Disclaimer.** Publisher's note: Copernicus Publications remains neutral with regard to jurisdictional claims in published maps and institutional affiliations.

**Acknowledgements.** This work would not be possible without the generosity of all the modelling groups that donated their simulation output and the Earth System Grid Federation for distributing all that output. This work was inspired by a Global Engagement award from University College London.

**Financial support.** This research has been supported by the Natural Environment Research Council (grant no. NE/S009736/1) and the Directorate for Geosciences (grant no. 1903482).

**Review statement.** This paper was edited by Qiuzhen Yin and reviewed by Mehdi Pasha Karami and two anonymous referees.

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
