# Peer review of "Indian Ocean variability changes in the Palaeoclimate Modelling Intercomparison Project"

_EGUsphere, 2022_

## Author Comment (AC1)

**Response to Reviewers comments on "Indian Ocean variability changes in the Palaeoclimate Model Intercomparison Project" by Brierley et al.**

**We would like to thank Dr Pasha Karami and two anonymous referees for their kind comments about our research. Overall, the three reviews agree that they were contained in our initial submission is worthy of publication in climate of the past after some relatively minor revisions. Below we respond to each individual comment.**

**Referee #1**

*The manuscript entitled "Indian Ocean variability changes in the Palaeoclimate Model Intercomparison Project by Chris Brierley et al. explored the changes in the tropical Indian Ocean simulated by the Palaeoclimate Model Intercomparison Project (PMIP4). The manuscript is well written, and the findings from this study have implications for model development and point to the importance of understanding the internal variability of the climate system. Hence, I recommend this study for publication in EGUsphere. But before publishing, I would like to mention a few points observed in this study. I hope that adding those points can improve the manuscript.*

*1. The introduction section of the manuscript gives an interesting description of the Indian Ocean Dipole and Indian Ocean Basin Mode. However, the authors didn't explain the reason for choosing these two modes when there are others like Subtropical Indian Ocean Dipole (SIOD) (Behera et al.,2001 ), IOD Modoki (Endo et al. 2015), and Indian Ocean Tripole (IOT, Zhang et al., 2020) exists. Hence, it is wise to add a few more sentences to the first paragraph of the introduction. Explain why you have considered IOD and IOBM so that readers will find continuity in reading.*

**We were motivated to focus on the IOD and the IOBM, as these are the only two modes in the Indian Ocean that are described in the IPCC's Assessment Report 6. We have adopted the primary definitions of that pair of modes as using in their Annnex IV (Cassou et al., 2021).**

**However, we had not explained that anywhere in the manuscript itself. We will alter the second paragraph of the introduction to explicit state this and to mention that there are other modes of variability within the Indian Ocean.**

2. The study discusses the precipitation and temperature variability over the Indian and African regions. But in many figures, the variability of these parameters is noticed over the Arabian Peninsula regions, particularly during the Last Interglacial period. Hence, it would be worth adding the descriptions related to Arabian Peninsula to the manuscript.

**That is an interesting observation. We will add some mention of the climate influences over the Arabian Peninsula in Section 3.**

3. In the figure caption.5, Check line 4 and correct the spelling of simulation.

**The figure caption will be corrected.**

**Referee #2**

*This manuscript presents simulated Indian Ocean variability in the latest PMIP4 multi-model ensembles. The periods investigated are two characterised by orbital forcing—midHolocene and lig127k, and two by substantial global cooling and warming—LGM and abrupt4xCO2. The responses*

*of the mean climate and three interannual modes (the IOD, IOBM and Nino-like mode) compared to the those during the pre-Industrial were studied. The results show quite diverse responses across experiments and models. The robust changes are only present in limited scenarios for not all modes, for instance, weakening of the IOD in the midHolocene and strengthening of Nino-like mode in the lig127k. The relationship between the mean state changes and climate variability changes and between ENSO amplitude and that of the Indian climate modes were examined. The authors reported that the relationship in model simulations is not always in agreement with proxy data, such as IOD-zonal SST gradient relationship.*

*The paper is nicely written and very informative. It clearly represents a large and substantial amount of effort and careful analysis. The results will be of value to the paleoclimate community by raising new questions. I have a few comments, mostly surrounding clarity of figures and presentation, that would improve the paper.*

**We would like to thank the Referee for their kind words, and we agree that the proposed edits would improve the manuscript.**

*1. My only major comment is for the definition of climate mode indices. All these modes are indicated by area-average SST anomalies of the Indian Ocean, often with overlapped regions (Sec 2.3). The Nino-like mode index is also derived in a similar way due to data availability. I am not sure if these indices are significantly different from one another. Maybe this needs to be tested within uncertainty. A big concern is that this can partly contribute to their insignificant changes in different scenarios. Another potentially useful test is the mean change in SST and rainfall patterns of IOBM and Nino-like mode (similar to Fig. 5), and see the patterns are distinguished between modes.*

**We had followed the approach used by the IPCC to define both the IOBM and IOD. They seem perfectly happy to treat them independently, so we do not feel that testing for independence is necessary in the present manuscript.**

**This is clearly not the case for the Nino-like mode, which is part of the reason that we had chosen to treat it differently in the layout of the text. We created a scatterplot showing the relationship between the changes in the IOD and Nino amplitude, which satisfied us that they were somewhat independent (it is the top panel in the figure below). Nonetheless, we had tried to introduce an element of caution throughout the text. We will explicitly mention this issue in sect. 4.3 of a revised manuscript.**

**We had not looked to see if the IOBM and the Nino-mode were acting independently. We have now created a similar scatterplot (the bottom panel in the figure below), which surprisingly does show a roughly 1-to-1 relationship. However, and pertinent to our conclusions, this relationship does not hold in the lig127k simulations. This can already be seen by the fact that the IOBM reduces in lig127k ensemble (manuscript Fig. 4), yet the Nino-like mode increases (manuscript Fig. 8). We will add a statement to encompass this finding.**

[Figure]

2. L215. This statement may not be accurate. In LGM simulations, the presence of continental ice-sheets at higher latitudes can also impact the tropical climate through the atmospheric circulations, at least for the tropical Pacific (e.g. Lee et al. 2015; Lu et al. 2016).

**We were aware of this possibility and had mentioned it on L250 through a citation to Jones et al (2018). We would mention this fact earlier in a revised manuscript along with the additional citations.**

*3. The direct comparison of a transient state of warming of abrupt4xCO2 (year 101-150) and an equilibrium state of cooling of LGM can be misleading, e.g. due to delayed response in deeper ocean and thus the ocean stratification. This difference should be pointed out when discussing the results.*

**We will now added a sentence to remind readers that that the abrupt4xCO2 simulations are not fully equilibrated and that further changes may be expected.**

*Fig. 3 Please give the meaning of black contour in the figure caption.*

**The ensemble mean of the respective piControl simulations is overlain as black contours (shown for every 2 mm/day until 12 mm/day). This would be added to the figure caption.**

*Fig. 4 and others Please describe how you calculate the vertical profiles, and the meaning of shorter dashed lines. For each dot, does its location on x-axis mean anything?*

**The distributions are computed using a kernel density estimation from the shown simulation values, as are the quartiles shown as short dashed and dotted lines (Waskom, 2021). The horizontal location of the dots has no meaning and has been chosen to allow all simulations to be seen. We do not feel that it is necessary to add this to every single caption, but apologise for not mentioning it anywhere.**

*Fig. 7 There is no black solid line here, and it should be in Fig. 6.*

**The sentence describing the black solid line will be moved to the caption of Fig. 6**

**Dr Pasha Karami (Referee #3)**

*The manuscript discusses different modes of Indian Ocean variability in relation to different climatic forcing simulated within PMIP4. I have few critical comments and some minor comments which are listed below. It is an interesting work and I think it should be published in Climate of the Past after the revision.*

**Thank you for the time that you have spent reviewing our research, and the insightful comments about it.**

*The first main comment I have is about the writing structure. The paper contains a lot of information and the reader can easily lose the track of what is changing what with the current structure. For instance if one does not know much about the IOBM and its changes, they should go back and forth in the paper to keep the track of the story. Moreover, the paper slowly deviates from the main focus (given in title/abstract) and conclusions scatter a bit throughout the paper. I suggest if the authors could re-structure the paper and text to ease the read a bit, for instance (just a suggestion):*

  i.  *start with the definition of the variability modes early in the paper*

  ii.  *then continue how these modes look in observations/reanalysis and discuss their impact in positive/negative phases if relevant*

  iii.  *comparison to pre-industrial simulations + discussion*

  iv.  *comparison to and difference/changes between the different forcing (both in mean state and variability) + discussion*

  v.  *potential impact of those changes in terms of temperature and precipitation (regression analysis) + discussion.*

  vi.  *Move section 4.3 for Nino-like mode to section 3.3; the connection to ENSO keeps coming in different parts of the paper, try to move all the ENSO-related story to*

*section 4.2; move the definition of the modes in page 7 to page 2 or 3 where you mentioned them first.*

**This would be an alternate approach to help navigate the reader through the research findings. And we can see that it has some advantages. We do not feel that would provide sufficient benefits to warrant the work required to undertake this restructuring. This is, in part, supported by the other two referees, who do not find that the current structure is problematic.**

*The periods/frequencies of the modes are not mentioned. Next to the changes in modes' amplitude, their periods (or number of occurrence per 100 years) and how they change in different climate, should be discussed.*

**As the reviewer has stated themselves, there is already a lot going on in this manuscript and we feel that adding further information is not going to help with the readability. However, we have now calculated the dominant period for each mode and simulation and provide that information in the associated code and data repository. We feel that the sole interesting finding is that the orbital forcing seems to increase the frequency of event in the EEIO region. This will be mentioned in Sect. 4.3. We would also expand the Code & Data Availability statement to be more explicit about the contents of the repository.**

*This is a question more than a comment. For the variability in the Pacific and Atlantic Oceans, it is common to use the EOF modes to represent the modes of variability. I wonder how this will be for the Indian Ocean? I am not asking for new analysis, but it will be helpful if you could cite relevant studies which discuss the Indian Ocean variability using EOF (if any). The advantage of using EOF is that you can see if the patterns and variance of dominant modes and how they vary for different climatic forcing.*

**This is a fair point, and we now mention the existence of EOF-based metrics in Sect. 2.3.**

**In previous work (Brierley, 2015) on a different ensemble, we have found that the ordering of the EOFs can vary between experiments (at least in the Pacific). That was in a small ensemble, and so was readily visible by eye. This possibility could cause serious problems when using the kind of large multi-model, multi-experiment dataset analysed here. Therefore, we had determined to use SST indices, which do not suffer from such issues.**

*I think a plot (maybe for appendix) showing the standard deviation of temperature among the ensemble members for each experiment will be helpful, but it is not necessary.*

**We are not quite sure what standard deviation is intended in this comment. However, we feel that this point shows that we did not provide sufficient signposting to the associated code and data. As well as the individual spatial fields and time series for each simulation, there is a single spreadsheet of key statistics for each simulation. We will increase the discussion of this information in the revised code and data availability statement.**

*Other comments:*

*Title: needs to be revised (e.g. Indian Ocean variability in PMIP4 simulations)*

**We need to revise the title, but not in the fashion suggested. Personally, we try to avoid acronyms in paper titles, as we do not think that it helps with openness. This manuscript also combines PMIP4 with PMIP3 simulations (to increase the sample size), so we cannot adopt the title suggested. We will need to change 'model' to 'modelling' as we have not expanded the acronym PMIP correctly.**

*Line 72: "...Indian Ocean variability." of what? SST variability?*

**We will add SST into this sentence to clarify the nature of the variability.**

*Line 80: "...capture the key processes..." like which processes?*

**We were envisaging the Cloud-radiation-SST and wind-evaporation-SST feedbacks, when writing the sentence. However, the reference cited is not appropriate to support that meaning. So we will remove the mention of the key processes from the sentence.**

*Line 122: I am not sure if point c is correct. "that" should be than.*

**We had presumed it was the case that the abrupt-4xco2 run was closer to equilibrium than the 1% run, but had not specifically looked at the data to check. We felt this would be easy to see through available data online such as with the ESMValTool. There doesn't actually seem to be a CMIP6 portal of precomputed variables for these kind of outputs (https://cmip-esmvaltool.dkrz.de/ isn't working at the moment).**

**We have therefore explored the statement using the UKESM1-0-LL model only, whose data was closest to hand. The figure below shows the global energy imbalance at the top of the atmosphere – and equilibrium being when that imbalance reaches zero. The imbalance in the final 50 years of the abrupt-4xco2 simulation is less than that in the 1pctco2 simulation, although not by as much as we had imagined. Perhaps more importantly the direction of the travels in the 1pctco2 simulation's net TOA flux shows that it is moving further away from equilibrium (an imbalance of 0) as the run progresses, rather than towards it. We hope this convinces the reviewer that our point (c) is correct. We will also fix the typo noted.**

[Figure]

*Line 200-213 and 230-247: hard to follow, I suggest to re-write (recalling main comment)*

**We will this paragraph to make it more explicit that our results do not fit with the standard pattern expected from the literature. This is due to our adopted analysis technique and is the reason that we do give scant discussion of rainfall teleconnection changes over India in results.**

*Line 268: "Having established..." I think such sentences would help in re-structuring (main comment).*

**We do not think restructuring the manuscript is beneficial (see above). We do appreciate the point about topic sentences at the start of a paragraph and believe that they are sufficiently implemented in the present work.**

*Line 287-288: composite analysis versus regression: I am not sure if I agree with that sentence, and both techniques should have similar main features. You could also use difference-correlation in this part of the analysis.*

**We agree that if the compositing was performed using a relative threshold (such as +/- 1 standard deviation), then it could also be used to look at teleconnection patterns. We would remove the comparative clause from the revised manuscript. We are not familiar with the difference-correlation technique, but will investigate its potential use for further work.**

*Line 360: "positive relationship" you mean positive correlation?*

**We really meant a positive slope arising from linear regression, although would also imply a positive correlation.**

*Figure 2: color bar scale needs to be changed to show the anomalies better for 4xCO2*

**We feel that current color bar scale adequately shows that the anomalies are generally between 4-5oC over most of the Indian Ocean and show little variation across the basin or year (as would be expected under a spatially and temporally homogenous forcing, such as $CO_2$).**

**References**

IPCC, 2021: Annex IV: Modes of Variability [Cassou, C., A. Cherchi, Y. Kosaka (eds.)]. In Climate Change 2021: The Physical Science Basis. Contribution of Working Group I to the Sixth Assessment Report of the Intergovernmental Panel on Climate Change [Masson-Delmotte, V., P. Zhai, A. Pirani, S.L. Connors, C. Péan, S. Berger, N. Caud, Y. Chen, L. Goldfarb, M.I. Gomis, M. Huang, K. Leitzell, E. Lonnoy, J.B.R. Matthews, T.K. Maycock, T. Waterfield, O. Yelekçi, R. Yu, and B. Zhou (eds.)]. Cambridge University Press, Cambridge, United Kingdom and New York, NY, USA, pp. 2153–2192, doi:10.1017/9781009157896.018

---

## Author Response (AR2)

Thank you for accepting our manuscript for publication. This version is nearly identical to the recently submitted one, except that we have added the DOIs to the references.

I am also unsure whether I have explicitly pointed out that elements of the research were funded by NERC through grant number NE/S009736/1